# Post-Genomic Methodologies and Preclinical Animal Models: Chances for the Translation of Cardioprotection to the Clinic

**DOI:** 10.3390/ijms20030514

**Published:** 2019-01-25

**Authors:** Lina Badimon, Guiomar Mendieta, Soumaya Ben-Aicha, Gemma Vilahur

**Affiliations:** 1Cardiovascular Program- ICCC, Research Institute-Hospital de la Santa Creu i Sant Pau, IIB-Sant Pau, 08025 Barcelona, Spain; lbadimon@santpau.cat (L.B.); guiomar.mendietabadimon@gmail.com (G.M.); SBenaicha@santpau.cat (S.B.-A.); 2Centro de Investigación Biomédica en Red Cardiovascular (CIBERCV) Instituto de Salud Carlos III, 28029 Madrid, Spain; 3Cardiovascular Research Chair, Universidad Autónoma Barcelona (UAB), 08025 Barcelona, Spain; 4Department of Cardiology, Hospital Clinic, 08036 Brcelona, Spain

**Keywords:** post-genomics, -omics, targets, ncRNA, chaperones, cardioprotection

## Abstract

Although many cardioprotective strategies have demonstrated benefits in animal models of myocardial infarction, they have failed to demonstrate cardioprotection in the clinical setting highlighting that new therapeutic target and treatment strategies aimed at reducing infarct size are urgently needed. Completion of the Human Genome Project in 2001 fostered the post-genomic research era with the consequent development of high-throughput “omics” platforms including transcriptomics, proteomics, and metabolomics. Implementation of these holistic approaches within the field of cardioprotection has enlarged our understanding of ischemia/reperfusion injury with each approach capturing a different angle of the global picture of the disease. It has also contributed to identify potential prognostic/diagnostic biomarkers and discover novel molecular therapeutic targets. In this latter regard, “omic” data analysis in the setting of ischemic conditioning has allowed depicting potential therapeutic candidates, including non-coding RNAs and molecular chaperones, amenable to pharmacological development. Such discoveries must be tested and validated in a relevant and reliable myocardial infarction animal model before moving towards the clinical setting. Moreover, efforts should also focus on integrating all “omic” datasets rather than working exclusively on a single “omic” approach. In the following manuscript, we will discuss the power of implementing “omic” approaches in preclinical animal models to identify novel molecular targets for cardioprotection of interest for drug development.

## 1. Introduction

The Human Genome Project started in 1990 with the aim of determining the sequence of the human genome and improving disease diagnosis and genetic susceptibility to particular diseases [1]. The completion of this project in 2001 [2] marked the beginning of the post-genomic era characterized by the development of several approaches to extract useful biological information from the genomic data. These approaches included the holistic analysis (omics) by high throughput technologies and bioinformatics of the genes actively transcribed (transcriptomics); the proteins expressed (proteomics); and the profile of metabolites released/synthesized (metabolomics). “Omic” platforms have become key tools to improve our understanding of the pathogenesis of ischemic heart disease, identify diagnostic/prognostic/predictive biomarkers, and identify new molecular targets with therapeutic potential. In this latter regard, analysis of the global cardiac gene/protein/metabolite expression profile of the ischemic heart with or without cardioprotection (mainly ischemic conditioning approaches) has allowed identifying novel potential pharmacological targets.

In this review, we will focus on the use of “omic” technologies in animal models with high translatability with the aim to discover new cardioprotective targets with potential for drug development.

## 2. Cardioprotection

Ischemic heart disease, including myocardial infarction, is the leading cause of mortality worldwide, being myocardial infarct size the main determinant of long-term mortality and chronic heart failure [3]. Reperfusion, initially by thrombolysis and subsequently by primary percutaneous coronary intervention (pPCI), is the most effective treatment to manage acute ST-elevation myocardial infarction (STEMI), markedly reducing mortality and morbidity. Reperfusion, however, induces necrotic and apoptotic injury to myocytes that were viable before reperfusion, a process called lethal reperfusion injury, which contributes to overall cardiac damage [4,5]. Over the past years, numerous cardioprotective approaches as an adjunct to reperfusion therapy have proven to be effective in the attenuation of the reperfusion insult [6] and in reducing infarct size in experimental animal models (e.g., beta-blockers, calcium-channel blockers, anti-erythropoietin, statins, glucose-insulin-potassium therapy, adenosine) [7,8,9,10]. However, as per today, there is no effective therapy on the market to limit myocardial damage, since none has demonstrated to improve patient’s clinical outcomes [11]. There is a need to identify new molecular targets with therapeutic potential capable of attenuating the deleterious mechanisms triggered upon myocardial ischemia and further reperfusion.

The discovery, 30 years ago, of the ischemic conditioning phenomena, has amplified our knowledge of the molecular basis of cell injury and survival mechanisms during myocardial ischemia and reperfusion [12]. Ischemic conditioning is a powerful endogenous cardioprotective response triggered by the exposure to a series of short intervals of ischemia and reperfusion. This maneuver has shown to release multiple mediators including adenosine, bradykinine, catecholamines, opioids, and lipid molecules suggested to trigger signaling pathways involved in cardioprotection by interacting with G-protein coupled receptors. Ischemic conditioning can be implemented at different sites (either locally in the coronary artery or remotely in an organ distant from the jeopardized myocardium) and at different timings (being pre-conditioning applied in a prophylactic approach before the ischemic insult; per-conditioning during persistent myocardial ischemia; and post-conditioning, applied at the time of reperfusion). Ischemic conditioning has constantly demonstrated in animal studies to make the heart resistant to lethal reperfusion injury, being ischemic pre-conditioning the most reliable and consistent form of myocardial protection against infarction [13,14]. There have been described two phases of cardiac protection (Figure 1). The first phase of protection follows immediately after the induced stimulus and lasts 2 to 3 h and involves the activation of early mediators [protein kinase C (PKC), tyrosine kinase, phosphatidylinositol-4,5-bisphosphate 3-kinase (PI3K)-Akt, mitogen/extracellular signal-regulated kinase (MEK)1/2-Erk1/2, and (Janus kinase) JAK]. In turn, these cardioprotective kinases activate transcription factors (such as STAT1/3, AP1, Nrf2, and HIF-1) resulting in the synthesis 12–24 h later (the second phase of protection) of downstream mediators [induced nitric oxide synthase (iNOS), heat shock proteins (HSP) and cyclooxygenase-2 (COX-2)]. These, in turn, protect the heart against infarction by acting on ‘end-effectors’ which persists up to 2–3 days after conditioning [15,16]. Although the signal transduction pathway involved and “end effectors” remain to be fully identified, it is believed that ischemic conditioning cadioprotective effects appear to be mediated through the inhibition of MPTP opening at reperfusion and modulation of the mitochondrial ATP-dependent K+ channels (mitoK+ATP) thereby preserving the mitochondrial function and consequently improving cell survival [17,18]. Yet, other forms of conditioning have been reported. As such, a pioneering work of Meerson and collaborators demonstrated, in 1973 [19], that exposure to high altitude hypoxia limited myocardial necrosis duet to acute or prolonged ischemia and prevented life-threatening arrhythmias, a phenomena called intermittent hypoxic preconditioning (IHC). The mechanisms involved in IHC are multiple and complex and most, but not all, are common to those studied in ischemic conditioning (for further details please refer to a recent and excellent review by Mallet et al.) [20]. On the other hand, experimental studies have also demonstrated that the myocardium can also be preconditioned by non-ischemic procedures, such as physical exercise [21]. In this regard, training has shown to enhance several cardioprotective factors including the anti-radical defense system, nitric oxide release, heat shock protein (HSP), and sphingosine-1-phosphate.

The discovery of these endogenous cardioprotective mechanisms has encouraged the exploration of new ways to protect the ischemic/reperfused heart. In this regard, “omic” analysis in the setting of ischemic conditioning has provided us with novel targets for cardioprotection amenable to pharmacological development.

## 3. Animal Models of Cardioprotection: Why Pre-Clinical Animal Models?

### 3.1. Rodents vs. Swine

There is no ideal animal model able to recapitulate the complexity that characterizes the clinical scenario of STEMI patients undergoing pPCI. Patients undergoing revascularization present a certain gene predisposition, are often middle-advanced age with various risk factors/co-morbidities and co-medications and with the presence of a culprit coronary atherosclerotic lesion that triggers thrombotic total coronary occlusion. However, despite the differences between the human clinical condition and animal models, experimentally induced myocardial-infarction (MI) models offer valuable tools for understanding the MI pathophysiology, discover novel molecular targets and assess novel therapeutic strategies.

Interestingly, consensus statements and recommendations have just been released for the measurement of cardiac physiology in mice and pigs [22]. Information on best practice exercise to plan and execute studies involving myocardial ischemia and infarction has also been generated [23]. The two most used animal models in the field of cardioprotection are rodents and swine. Yet, other animal models including zebrafish, rabbits, dogs, sheep and non-human primates, have also been used in cardiac research and to test new cardioprotective compounds [24,25]. Yet, one should pay attention, however, to the weaknesses and strengths of rodents and swine, which are summarized in Table 1.

In the past, dogs were extensively used as animal models to study myocardial ischemia. However, canines demonstrate significant variations in the coronary circulation pattern between breeds (mongrels or purebreds) and have an extensive collateral circulation able to supply as much as 40% of the blood flow after coronary occlusion, markedly limiting myocardial damage [26]. Moreover, in dogs, in contrast to humans and swine, the atrioventricular node and the bundle of His are irrigated predominantly by the anterior and not by the posterior septal artery. This implies that injury to the conduction system in the canine differs from that of humans and pigs [27,28]. As a whole, because porcine heart bears a close resemblance to the human heart in terms of its coronary circulation, cardiac structure and hemodynamics, they are considered to have the greatest translational potential in the field of cardioprotection. As such, the ESC-Working Group Cellular Biology of the Heart Position Paper highlighted that study in pigs is an obligatory step before initiating human trials in cardioprotection [29]. Indeed, pig heart differs from humans in its anatomic orientation in the thoracic cavity. These traits, however, are common to most mammals used in biomedical research [30].

### 3.2. Important Factors to Consider When Performing Cardioprotective Studies in Pre-Clinical Animal Models

In addition to the existing differences between animal models, there are important factors that must be considered when performing pre-clinical studies because they can certainly impact the “omic”-related findings. These are detailed in Figure 2. Among all the factors depicted in the figure, special attention should be given to the impact of confounding factors since they have shown to attenuate or abolish the cardioprotective effect of ischemic conditioning [31,32]. Gender has shown to impact on ischemia/reperfusion (I/R) injury and ischemic conditioning-related cardioprotection [33]. Female hearts have shown to display a higher resistance to I/R injury likely because a different distribution of PKC and ERK isoforms compared with male hearts [34]. Moreover, the considerable tolerance of the female heart can be further improved by ischemic preconditioning, although this protective effect seems to be age-dependent thereby decreasing with ageing. In line with this sex/age-related observation, several studies have reported a loss of ischemic conditioning-related cardioprotection in aged hearts likely associated with lack of Akt activation and/or failure to prevent MPTP opening. Another important confounding factor is the oxygen supply/demand ratio since an imbalance between myocardial oxygen supply and demand (e.g., decreasing cardiac oxygen consumption by controlling heart rate and hemodynamics) may limit the size of infarction masking any conditioning-related effect. Likewise, multiple animal and human studies have repeatedly demonstrated that both short-term and long-term exercise training decrease ischemia reperfusion-induced cardiac injury through a pre-conditioning effect [35]. Although the mechanisms responsible for exercise-induced cardioprotection remain elusive, it is considered that several substances and metabolites released during exercise including lactate, adenosine, bradykinin, and opioids may activate the downstream pre-conditioning kinase cascade. Moreover, exercise may also induce pre-conditioning by directly causing myocardial hypoxia or by nitric oxide induction. Finally, as per the impact of cardiovascular risk factors, in addition to hyperlipidemia, most of the other major risk factors including hypertension, obesity, metabolic syndrome, diabetes, and uremia have shown to modify the cardioprotective signaling involved in ischemic conditioning thereby attenuating its protective effects [36,37].

Overall, for identification of valid drug targets, the effect of cardiovascular risk factors, co-morbidities, and co-medications of ischemic heart disease should be taken into account [31], and the novel drug target should still be active in the presence of major cardiovascular co-morbidities and their routine medications.

## 4. Implementation of Post-Genomic Technologies in Preclinical Animal Models to Identify New Targets/Strategies for Cardioprotection

### 4.1. Transcriptomics

The past 15 years have witnessed an enormous rise in the development of high-throughput porcine transcriptomic data. Porcine expressed sequence tags, serial analysis of gene expression [38] and gene arrays have expanded greatly because of biomedical interest, particularly in the context of pig immunity [39] and for the analysis of muscle phenotypes [40]. In fact, quantitative real-time PCR assays to validate the differential expression of porcine transcript have become publicly available. Unfortunately, however, pig transcriptomic studies have been scarcely conducted in the cardiovascular system and at most, interest has focused on the systematic analysis of cardiovascular-related non protein-coding RNAs, including microRNAs (miRNAs or miR; 21–25 nucleotides in length) and long-noncoding RNAs (lncRNA; >200 nucleotides), that regulate gene expression at a post-transcriptional level without modifying the DNA sequence. miRNAs have been extensively investigated in the setting of acute MI, and several studies have demonstrated early changes in the cardiac expression of miRNAs (either increasing or decreasing) in response to I/R. Moreover, pharmacological manipulation of these miRNAs has been shown to modulate the sensitivity of the myocardium to acute I/R thereby implicating miRNAs as promising novel targets for cardioprotection [41,42]. So far, changes in myocardial expression of several miRNAs have been reported in ischemic conditioning although only a few have demonstrated, in rodent hearts, to contribute to ischemic conditioning-related cardioprotection directly (Table 2) [42].

Regarding lncRNAs, although most of the transcripts are linear, a circular type of lncRNA has been identified [50]. In cardiovascular pathology the circRNA MFACR (mitochondrial fission and apoptosis-related circRNA) has been found to mediate cardiovascular death in a mouse model of I/R by directly targeting and downregulating miR-652-3p, which in turn, blocks mitochondrial fission and cardiomyocyte cell death by suppressing the translation of a mitochondrial protein involved in cell survival (MTP18) [51].

### 4.2. Proteomics and Post-Translational Modifications

#### 4.2.1. Proteomics

Any cell expresses all genes simultaneously; yet, there is selective gene transcription depending on the cell type and the stimuli to which it is exposed. Proteomics allows the characterization of those proteins responsible for a cell function at a given moment. Elmadhun and colleagues examined, by proteomic analysis, the effects of daily moderate alcohol consumption in the pericardium and myocardium in a swine model of chronic myocardial ischemia [52]. The authors reported that alcohol intake increased structural proteins, and decreased immune protease inhibitors and coagulation proteins in the pericardium and decreased remodeling proteins, cell death proteins and motor proteins, and increased metabolic proteins in the myocardium. These data suggest that daily moderate alcohol consumption affects numerous pathways that contribute to cardioprotection, including cardiac remodeling, metabolism, and cell death. Our group has identified, by implementing post-ischemic conditioning approaches, various molecular chaperones like heat shock proteins (HSP) as potential therapeutic targets to protect against MI-related damage and prevent cardiac remodeling post-MI. HSP acutely respond to sudden stress playing a vital role in helping proteins to become properly folded or in aiding misfolded proteins to regain their correct conformation making chaperones an important target in cardiovascular diseases [53]. We demonstrated in pigs that myocardial damage due to acute MI is associated with loss of protective chaperones. Furthermore, by implementing proteomic approaches, we have observed a differential distribution profile of Hsp70 and Hsp90, best-known as high molecular weight chaperones with a coordinated action in the myocardium after ischemic post-conditioning in comparison to immediate full reperfusion [54]. Moreover, we have also identified that ischemic post-conditioning preserves myocardial expression of the cardioprotective small HSPs alpha-crystallin-B-chain (Cryab/HspB5) which is found markedly diminished due to I/R. Most importantly, Cryab/HspB5 protein levels were associated with improvement of cardiac function and infusion of Cryab/HspB5 prior MI protected against myocardial ischemic damage [55]. Altogether our observations support a potential cardioprotective role for Cryab/HspB5. By using proteomic approaches in pigs, we have also recently characterized the composition-related changes that suffer high-density lipoproteins (HDL) formed under hypercholesterolemic conditions that render them dysfunctional to exert their cardioprotective effects [56,57]. As such, those HDL particles formed and remodeled in a low-density-lipoprotein cholesterol-rich milieu carry a lower cargo of proteins with cardioprotective properties including lipocalins retinol binding protein (RBP)-4, Apo-M and the retinoic acid-transporter CRAPB-1 [58].

#### 4.2.2. Post-Translational Modifications

Post-translational modifications can regulate the myocardial proteome. Various proteins and signaling pathways orchestrating cardioprotection are mainly regulated by phosphorylation and to a lesser extent by O-GlcNAcylation [59].

##### Phosphoproteomics

Phosphoproteomics identifies proteins containing a phosphate group as a post-translational modification whose activation or inhibition regulates cell survival, apoptosis, calcium homeostasis, and reactive oxygen species [60]. Please note that whereas kinases regulate phosphate attachment to serine, threonine, and tyrosine residues, phosphatases remove the phosphate from the protein. PI3K activation is required for phosphorylation of Akt (pAkt), a pro-cell survival kinase. P-Akt, in turn, opens the mitoK+ATP stabilizing potassium levels and maintaining membrane potential, and also phosphorylates Bcl-2 associated death promoter causing its translocation from Bcl-2, thus preventing cell death due to apoptosis [61]. Another key signaling pathway in cardioprotection involves the phosphorylation of JAK, which subsequently leads to the activation of signal transducer and activator of transcription 3 (STAT3) and the phosphorylation of PKCε preventing the opening of the mPTP [62]. Gedig et al. compared the phosphoproteome of left ventricle biopsies from pigs undergoing coronary occlusion/reperfusion without and with remote ischemic conditioning. They detected that phosphorylation of 116 proteins differed between both groups of animals and those proteins found to have a higher degree of phosphorylation (≥2-fold) in conditioned animals were related with the mitochondria and cytoskeleton [63].

##### O-GlcNAcylation

Research into the role of GlcNAcylation in post-translational modifications is relatively new. The O-linked attachment of β-N acetylglucosamine (O-GlcNAc) has shown to regulate cardioprotective proteins involved in calcium homeostasis and reducing apoptosis [64]. Recent studies in pigs have demonstrated that an acute increase in OGlcNAc levels, either prior ischemia or at reperfusion, limits the size of infarction and improves heart recovery [65,66].

### 4.3. Metabolomics and Lipidomics

So far, few studies have addressed the metabolomic and lipidomic changes associated with cardioprotection. The metabolic profile provides readout of the metabolic state of a specific physiological/pathogenic situation [67]. An interesting study in pigs subjected to complete coronary balloon occlusion described a metabolomic biosignature in blood serum associated with myocardial ischemia which included metabolites associated with myocardial energy production and lipolysis activation in the adipose tissue. Interestingly, this metabolomic profile was further validated in a proof-of-concept study in patients undergoing transitory myocardial ischemia in the setting of planned coronary angioplasty [68]. Ischemic conditioning has also shown to modulate in experimental animal model at both tissue and blood levels, metabolites involved in glycolysis, glutathione oxidation balance, synthesis of glycogen and amino acids, as well as fatty-acids and sphingolipid metabolism [69,70]. A recent study using untargeted metabolomic analysis of plasma, following remote ischemic conditioning in swine, revealed a mild impact of the cardioprotective maneuver on the plasma metabolome [71]. At a tissue level, implementation of two models of delayed preconditioning demonstrated metabolic signatures completely different. As such, the frequency and magnitude of the metabolite changes was higher in those animals subjected to repetitive coronary stenosis (6 episodes of non-lethal ischemia over 3 days prior to the ischemic insult) as compared to those that underwent second window of ischemic preconditioning (two episodes of total left anterior descending coronary artery occlusion for 10 min and 10 min of reperfusion, followed by 24 h of reperfusion). Most interestingly, most of the altered cardiac metabolites found in repetitive coronary stenosis were linked to mechanisms known to improve energy metabolism [72]. Regarding lipidomics, we have recently reported that the attenuation of HDL-related cardioprotection in the setting of hypercholesterolemia might be associated with lipidomic changes in the HDL micelles, losing phosphatidylcholine-lipid species (involved in HDL functionality) and gaining cholesteryl esters (involved in HDL maturation and loss of function) [58].

### 4.4. Epigenetic Modifications: Methylation and Acetylation

Finally, cardioprotection research has also to integrate epigenetics. Epigenetics refers to the alteration of gene expression without modifying the DNA sequence. In this sense, genes may suffer modifications of their DNA (and histones) through methylation or acetylation eventually modulating gene activity. Methylation involves adding a methyl group in the DNA thereby preserving DNA positive charge and pronouncing its curvature leading to diminished gene transcription. In contrast, the addition of an acetyl group (acetylation) neutralizes DNA charge and lessens DNA bending and further transcription. The addition of an acetyl group is carried out by histone acetyltransferases whereas its removal by histone deacetylases (HDACs). So far, only one study in swine hearts has revealed modifications in the methylation of myosin light chain isoforms in the left ventricle as compared to the atria paving the way for a better understanding of their functional roles in cardiac physiology and pathophysiology [73]. On the other hand, many HDACs have been identified with impacts on cardiac cell function, and they have been classified from class I to V according to their function. Sirtuins (SIRT, SIRT1-7) from HDAC class III are gaining popularity in the field of cardioprotection research due to their impact on aging, apoptosis, and stress responses; however, they cannot act alone. As such recently SIRTs in combination with forkhead box containing protein O family (FOXO) have been identified as potential modulators of interest within the cardioprotection research [74].

## 5. Conclusions and Future Perspectives

It is clear that genomic information alone, although crucial, is not sufficient to explain the complexity that surrounds ischemic heart disease. The development of post-genomic approaches has allowed widening our understanding of MI-injury with each approach capturing a different angle of the global picture of disease. Transcriptomics has revealed changes in non-coding RNA in the setting of I/R thereby identifying new players in the modulation of cardiac injury; proteomics has determined alterations in protein expression and post-translational modifications upon implementing an endogenous form of cardioprotection called ischemic conditioning, thereby providing us with therapeutic candidates amenable to pharmacological exploitation; and, metabolomics and lipidomics are emerging as important tools to depict changes in both global and cardiac-specific metabolic profiles that occur in ischemic heart disease. However, still much work needs to be done to make sense out of all the available “omic” data to take the next step in overcoming pharmacological deficits to protect the heart from MI. Indeed, the discovery of potential therapeutic targets for cardioprotection requires insight into transcript/protein/metabolite interaction. Accordingly, efforts should focus on integrating all “omic” datasets rather than working exclusively on a single “omic” approach [75]. Integrated “omic” approaches with the subsequent robust network analysis will identify key regulatory networks and signaling hubs in response to cardioprotection that may lead to drug innovation [31]. It is important, however, to bear in mind that the progress made in drug discovery will only reach the patient when sound and reliable pre-clinical proof-of-concept data are available.

## Figures and Tables

**Figure 1 ijms-20-00514-f001:**
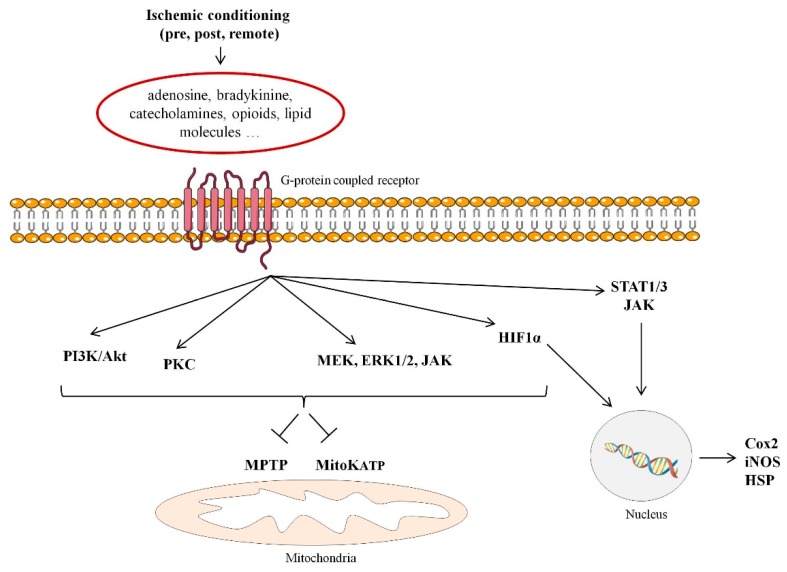
Signaling pathways involved in ischemic conditioning-induced cardioprotection. MPTP: mitochondrial permeability transition pore; mitoKATP: mitochondrial adenosine triphosphate-dependent potassium; PKC: protein kinase C; PI3K: phosphatidylinositol-4,5-bisphosphate 3-kinase (PI3K); MEK1/2: mitogen/extracellular signal-regulated kinase; JAK: Janus kinase; STAT1/3: signal transducer and activator of transcription; 1 HIF-1α: Hypoxia-inducible factor 1-alpha; iNOS: induced nitric oxide synthase; HSP: heat shock proteins; Cox2: cyclooxygenase-2.

**Figure 2 ijms-20-00514-f002:**
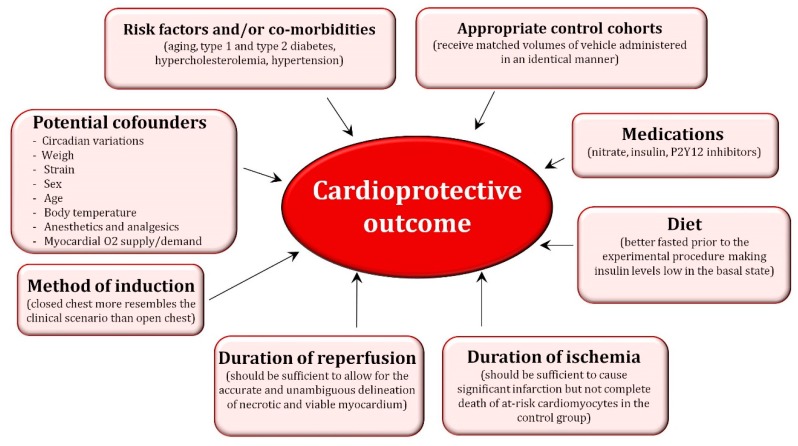
Factors that may contribute to the cardioprotective outcome.

**Table 1 ijms-20-00514-t001:** Strengths and weaknesses for conducting cardioprotection-related studies in the most commonly used animal models of myocardial infarction.

Animal	Strengths	Weaknesses
**Rodents**	Low cost, readily available, easy handling and housing, short gestation time, and low maintenance costs Genomic manipulation (knock-out and knock-in) Small size which limits the quantities of new agents required for in vivo screening of new cardioprotective compounds Availability of biological information for research purposes (databases, antibodies) Reduced ethical concerns	Cardiac anatomical/physiological differences from humans: largely differ in the coronary and myocardial architecture, heart rate (an order of magnitude higher than that of humans), oxygen consumption, and contractility. Limited size of infarction (no more than 70% of the area-at-risk). Despite permanent coronary occlusion, the inner myocardial layers continue to be served with oxygen and nutrients by diffusion. Atypical geometry of non-transmural infarcts. The subendocardium is spared from death by diffusion of oxygen from the left ventricular cavity and occupies an excessive proportion of the total left ventricular wall thickness. Inherent variability and the need for a large sample size. Transgenic mice models may present compensating mechanisms and redundancies that may affect the cardioprotective approach. Their size hampers the implementation of clinical surgical methods.
**Swine**	The heart is anatomically comparable to humans as well as both function of the coronary system (the left coronary artery supplies the majority of the myocardium) and the histological anatomy of the aorta. The heart has a sparse network of collaterals and a poor ability to recruit new collaterals during an acute ischemic event. Hemodynamic values are analogous to those in humans. Infarct size upon balloon occlusion of the left anterior descending coronary artery can be precisely predicted. Infarction is total and damages more than 80% of the area-at-risk. The ratio of heart weight to body weight in 30-kg pigs is identical (5 g/kg) to that of adult humans. Allows the use of the same interventional devices used in the clinical setting.	Substantial cost, require surgical facilities, specialized equipment, and expert personnel, limiting the feasibility of the procedure. Tolerance to ischemia/reperfusion varies notably across different pig strains. Susceptible to lethal ventricular arrhythmias. High growth rate and adult size of pigs may present a husbandry challenge for standard laboratory facilities.

**Table 2 ijms-20-00514-t002:** miRNAs reported to contribute to ischemic conditioning.

Mice	microRNA	Associated Function	Reference
Pre-conditioning	miR21	Related with apoptosis	[43]
	miR451	Related with oxidative stress	[44]
Post-conditioning	miR21	Related with apoptosis	[45]
	miR499	Related with apoptosis	[46]
**Swine**			
Post-conditioning	miR29b	Muscle specific	[47]
	miR133a	Related with fibrosis	[47]
	miR146b	Related with inflammation	[47]
	miR92a	Related with angiogenesis	[48,49]

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
