# Peer review of "Post-Genomic Methodologies and Preclinical Animal Models: Chances for the Translation of Cardioprotection to the Clinic"

_ijms, 2019, doi:10.3390/ijms20030514_

Round 1
Reviewer 1 Report
I don't have concerns.
Reviewer 2 Report
I thank the authors for making the changes in their manuscript. I also noted improvements in other sections (mRNA's e.g.) that make the manuscript a nicer overview.
Thank you for explaining the effectors towards the mitochondrial K ATP channel, and Fig 1 supports the text well.
This manuscript is a resubmission of an earlier submission. The following is a list of the peer review reports and author responses from that submission.
Round 1
Reviewer 1 Report
The review paper by Badimon et al concerns the pre-clinical animal models to be used in the omics approach to cardioprotection. The review is of interest because the post-genomic era is posing challenges that have not been totally addressed. Here a few suggestions to improve its quality.
The manuscript is slightly unbalanced in the section on metabolomics/lipidomics, which needs to be expanded, and that on proteomics, which looks excessive in self-citation.
The paragraph on cardioprotection (lines 66-88) misses important preconditioning issues, as hypoxic preconditioning (e.g., intermittent hypoxia), remote conditioning and physical training. It is questionable whether closing MPTP (line 86) is the main way to precondition hearts.
In Lines 95 and following, although rodents and swine surely represent good models to recapitulate cardiovascular diseases, other emerging models as zebrafish need be cited.
Table 1. I wonder whether rodent weakness #2 (limited infarct size) is really a weakness, because MI size in humans may not be so high in order to be still treatable. Also, the issue on age, sex and strain that influence MI size is a weakness for the swine model too. I can’t understand the swine “strength” related to the same HW/BW ratio as in humans, please clarify. Can swine be moderately trained as rodents, for example by placing a wheel in the cage? Finally, considerations of the male vs. female differences are lacking.
Line 170 and Figure 1. I appreciate the considerations on the confounding factors that may be so critical as to rule out strengths and weaknesses of the animal models. This section may require a full paragraph, especially the issue on myocardial O2 supply/demand, which may be very critical. Other important related issues are male vs. female, age, co-morbidities, degree of training and previous exposure to hypoxia-related stresses.
The conclusion statement, which appears weak, needs to be realigned and better finalized.
Specific issues.
Line 30 “limitations that can’t be overlooked” needs to be rephrased.
Lines 198-209 may be clearer in a Table. Also 226-242.
Author Response
REVIEWER #1: We thank the Reviewer for his comments which have helped to improve the manuscript. We have provided a point-by-point answer to the Reviewer comments below. Changes are highlighted in red for tracking purposes in the new version of the manuscript.
1.- The manuscript is slightly unbalanced in the section on metabolomics/lipidomics, which needs to be expanded, and that on proteomics, which looks excessive in self-citation.
We are aware of such imbalance. However, there is no much preclinical data available in the field of cardioprotection with regards to metabolomics and lipidomics. After another intense research in PubMed, we now comment on studies focused on the lipidomic profile of the pig heart as well on one study in pigs and humans aimed at identifying a metabolomic signature associated to myocardial ischemia. In addition, we now provide further details regarding the metabolomic changes described in the different settings of ischemic conditioning to further extend this section.
2.- The paragraph on cardioprotection (lines 66-88) misses important preconditioning issues, as hypoxic preconditioning (e.g., intermittent hypoxia), remote conditioning and physical training. It is questionable whether closing MPTP (line 86) is the main way to precondition hearts.
Following the Reviewer’s indication, we now include the suggested preconditioning concepts, toned down the impact of closing MPTP in conditioning-related cardioprotection and also added the key role of mitochondrial ATP-dependent K+ channels (mitoK+ATP) as “end-effector”.
3.- In Lines 95 and following, although rodents and swine surely represent good models to recapitulate cardiovascular diseases, other emerging models as zebrafish need be cited.
Following the Reviewer indication, we now mention zebrafish, rabbits, dogs, sheep and non-human primates as other animal models used in cardiac research and to test new cardioprotective compounds.
4.- Table 1. I wonder whether rodent weakness #2 (limited infarct size) is really a weakness, because MI size in humans may not be so high in order to be still treatable. Also, the issue of age, sex and strain that influence MI size is a weakness for the swine model too. I can’t understand the swine “strength” related to the same HW/BW ratio as in humans, please clarify. Can swine be moderately trained as rodents, for example by placing a wheel in the cage?
- Within cardioprotection, small infarcts represent a challenge to accurately depict the cardioprotective potential associated with a new approach or compound. Large infarcts facilitate the detection of cardioprotection.
- Regarding the impact of age, sex and strain, we completely agree that they also affect swine and have been accordingly removed this concept from the table and placed it in the figure.
- Comparable HW/BW ratio between swine and humans allows to more reliably translate the pig findings into the human scenario. As such, the growth of the heart and cardiovascular system from birth to 4-5 months of age (used experimentally) is analogous to the growth of the same system in humans into the mid-teens (Swindle MM, et al Comparative anatomy and physiology of the pig. Scand J Lab Anim Sci. 1998;25(suppl 1):1–10. Swindle MM. Swine in the Laboratory: Surgery, Anesthesia, Imaging and Experimental Techniques. 2nd ed. Boca Raton, FL: CRC Press; 2007).
- Finally, pigs can indeed be treadmill-trained as rodents.
5.- Line 170 and Figure 1. I appreciate the considerations on the confounding factors that may be so critical as to rule out strengths and weaknesses of the animal models. This section may require a full paragraph, especially the issue on myocardial O2 supply/demand, which may be very critical. Other important related issues are male vs. female, age, co-morbidities, the degree of training and previous exposure to hypoxia-related stresses.
We now provide an entire page dedicated to discuss the impact of the confounding factors suggested by the Reviewer.
6.- The conclusion statement, which appears weak, needs to be realigned and better finalized.
We have improved the conclusion section according to the reviewer’s comment.
Specific issues
1.- Line 30 “limitations that can’t be overlooked” needs to be rephrased.
The sentence has now been rephrased.
2.- Lines 198-209 may be clearer in a Table. Also 226-242.
- The concepts provided in Lines 198-209 have now been expanded and are depicted in Figure 1.
- Following the reviewer suggestion, the information provided in lines 226-242 is now included in a new Table 2.
Reviewer 2 Report
The paper by Badimon, Medieta, Ben-Aicha and Vilahur is a review on recent advances in our understanding of ischemic conditioning and at the same time a "line-up" for the next steps in research on this subject. The broader idea is that technological improvements in automated analytics (be it in genome, transcriptome or proteome, or even with plasma and tissue metabolomics and lipidomics) lead to more insight in activated pathways for ischemic conditioning, but that we still have to make sense of all these findings before therapeutic actions are taken on single findings. As such, the paper is a warning for investigators, and a call for action on integrating the recent findings. In the paper also a commentary is written (lines 155-168) about the best animal model to investigate ischemic conditioning, with reference 24 to back up, but the arguments in the text are merely anatomical (collateral vasculature, number of coronary arteries), while the arguments in the table 1 are more diverse; still, the authors are not keen on excluding any information coming from other models, so I regard this information as extra, but not as central.
Comments
1. The title captures the message on "post-genomic methodologies", but does not capture that findings from these methodologies should be integrated before therapies are started, it only has: "Innovating pharmacology in cardioprotection" . However, the paper is not on innovating pharmacology, it warns against too rapid development in this area without the proper research or understanding. So, maybe the authors can puzzle about what should be the proper title, for example: "Post-genomic methodologies for and understanding of ischemic conditioning", or something like the Lancet review by Heusch in 2013 on the subject of cardioprotection: "challenges of its translation to the clinic". The message of integrating all this research, as in the conclusion, is missing in the abstract, with perhaps too much attention for the animal models of choice in the abstract.
2. It is a little bit strange to see the same 'Intro sentence" in lines 32-33 in the abstract and 47-49 in the introduction, maybe you can skip the lines 32-33.
3. In introduction the phases of ischemic conditioning are described (lines 76-88), but since this information is quite essential for further reading, the reader (such as myself) has to go back to other papers (and figures) (refs 14-16) to understand what was said: the stimuli are not well described, and the early and downstream mediators involved are summarized without context (pathways involved, intracellular, intranuclear, etc). Then the "end-effectors" are also placed between hyphens, obviously quoting the original Cohen and Downey terminology; in Br J Pharmacology 2015 a historical perspective was published by them), with the end-effector in a sentence later determined as the mitochondrial permeability transition pore (ref 17, 18). This is a little bit vague, maybe there are other 'end- effectors" ?. You may consider to add a smal figure to the manuscript in which this is made better visible, and in which you also add the extra information described in the section of proteomics. In this way, it will be better highlighted what the progress in 'omics' has been. Your molecular chaperonnes like HSP are not placed in a known pathway now, or even with some sense of where they would do their work preventing ischemia/reperfusion induced damage, or what they then prevent (playing a role in helping proteins to become properly folded etc" line 225 makes me wonder which proteins and where they have their main actions). Perhaps you can make this sentence more clear, as it is your own research, and in the proposed figure locate this part of the protection system as 'new'evidence.
4. In the transcriptomics discussion (section 4.1), the microRNA's are discussed, but the presently investigated circular RNA's are not yet discussed. I have seen papers on ischemic conditioning and circular RNA's in other organs than the heart, but I am not an expert, and perhaps the paper of Holdt et al ( Molecular functions and specific roles of CircRNAs in the cardiovascular system, in: Non-coding RNA research 2018;3: 75-98) may help you further. I think this would add to the present knowledge and methodologies.
5. There used to be a lot of information on ATP-dependent K+ channels in conditioning; which is now in resume in paragraph 4.2.2.1 on phoshophoproteomics. Previously it was known that activation of such channels (with nicorandil) could increase the conditioning effect. Your statement in line 254-55, that this ultimately prevents cell-death by uncoupling of a 'death promotor' of Bcl-2 (line 255), is that the sole explanation for this ATP-dependent K channel activation dependent improvement in conditioning ? In this case, you already have a channel that works in the mitochondrial membrane, quite far from the other 'mediators', but you may also state quite close to the 'end-effector' and maybe you can explain why this would not be the target for pharmacological intervention in conditioning ? Is this a reasoning that would help future research, to get as close to the 'end-effector' in preventing reperfusion damage, or would you state that mediators and stimuli are as important ?
6. In section 4.3 on metabolomics and lipidomics you may have hurried through the evidence with 3 papers (57, 58, 45). I can see the value of metabolomics and lipidomics as an understanding around the pathways involved for conditioning, but are 'mechanisms known to improve energy metabolism' (line 279-80) other mechanisms? Please clarify whether the known conditioning pathways are already understood better from this perspective or whether no attempt has been done to do so.
Author Response
REVIEWER #2: We are very grateful with the Reviewer for providing us with these pertinent comments. A point-by-point answer to the reviewer comments is provided below and the manuscript has been revised accordingly. Changes are highlighted in red.
1. The title captures the message on "post-genomic methodologies", but does not capture that findings from these methodologies should be integrated before therapies are started, it only has: "Innovating pharmacology in cardioprotection”. However, the paper is not on innovating pharmacology, it warns against too rapid development in this area without the proper research or understanding. So, maybe the authors can puzzle about what should be the proper title, for example, "Post-genomic methodologies for and understanding of ischemic conditioning", or something like the Lancet review by Heusch in 2013 on the subject of cardioprotection: "challenges of its translation to the clinic". The message of integrating all this research, as in the conclusion, is missing in the abstract, with perhaps too much attention for the animal models of choice in the abstract.
Following the Reviewer suggestion, we now provide a new title that more accurately reflects the content of the review and have also amended the abstract according to her/his comment.
2. It is a little bit strange to see the same 'Intro sentence" in lines 32-33 in the abstract and 47-49 in the introduction, maybe you can skip the lines 32-33.
We consider that this sentence summarizes the content of the review and accordingly should be included within the abstract, which is displayed in PubMed, and as expected within the review. However, we have rephrased the sentence within the main text to avoid redundancies.
3. In introduction the phases of ischemic conditioning are described (lines 76-88), but since this information is quite essential for further reading, the reader (such as myself) has to go back to other papers (and figures) (refs 14-16) to understand what was said: the stimuli are not well described, and the early and downstream mediators involved are summarized without context (pathways involved, intracellular, intranuclear, etc). Then the "end-effectors" are also placed between hyphens, obviously quoting the original Cohen and Downey terminology; in Br J Pharmacology 2015 a historical perspective was published by them), with the end-effector in a sentence later determined as the mitochondrial permeability transition pore (ref 17, 18). This is a little bit vague, maybe there are other 'end- effectors" ?. You may consider to add a smal figure to the manuscript in which this is made better visible, and in which you also add the extra information described in the section of proteomics. In this way, it will be better highlighted what the progress in 'omics' has been. Your molecular chaperonnes like HSP are not placed in a known pathway now, or even with some sense of where they would do their work preventing ischemia/reperfusion induced damage, or what they then prevent (playing a role in helping proteins to become properly folded etc" line 225 makes me wonder which proteins and where they have their main actions). Perhaps you can make this sentence more clear, as it is your own research, and in the proposed figure locate this part of the protection system as 'new'evidence.
We understand the Reviewer’s point and in addition to extending our explanation on ischemic conditioning, we have created a new figure according to his/her suggestions.
4. In the transcriptomics discussion (section 4.1), the microRNA's are discussed, but the presently investigated circular RNA's are not yet discussed. I have seen papers on ischemic conditioning and circular RNA's in other organs than the heart, but I am not an expert, and perhaps the paper of Holdt et al ( Molecular functions and specific roles of CircRNAs in the cardiovascular system, in: Non-coding RNA research 2018;3: 75-98) may help you further. I think this would add to the present knowledge and methodologies.
We thank the reviewer for bringing this important and novel issue with its pertinent reference. We have now discussed circular RNAs within the transcriptomic section.
5. There used to be a lot of information on ATP-dependent K+ channels in conditioning; which is now in resume in paragraph 4.2.2.1 on phoshophoproteomics. Previously it was known that activation of such channels (with nicorandil) could increase the conditioning effect. Your statement in line 254-55, that this ultimately prevents cell-death by uncoupling of a 'death promotor' of Bcl-2 (line 255), is that the sole explanation for this ATP-dependent K channel activation dependent improvement in conditioning ? In this case, you already have a channel that works in the mitochondrial membrane, quite far from the other 'mediators', but you may also state quite close to the 'end-effector' and maybe you can explain why this would not be the target for pharmacological intervention in conditioning ? Is this a reasoning that would help future research, to get as close to the 'end-effector' in preventing reperfusion damage, or would you state that mediators and stimuli are as important ?
We apologize because a “coma” was missing in the sentence thereby conveying a misleading message.
As such, Akt is the one ultimately involved in Bcl2 phosphorylation, not mitoK+ATP. Now reads, “P-Akt, in turn, opens the mitoK+ATP stabilizing potassium levels and maintaining a membrane potential, and also phosphorylates Bcl-2- associated death promoter causing its translocation from Bcl-2, thus preventing cell death due to apoptosis”.
6. In section 4.3 on metabolomics and lipidomics, you may have hurried through the evidence with 3 papers (57, 58, 45). I can see the value of metabolomics and lipidomics as an understanding of the pathways involved for conditioning, but are 'mechanisms known to improve energy metabolism' (line 279-80) other mechanisms? Please clarify whether the known conditioning pathways are already understood better from this perspective or whether no attempt has been done to do so.
Unfortunately, there is no much preclinical data available in PubMed as regards to metabolomics and lipidomics in cardioprotection. However, we now provide a couple of studies that further support the importance of conditioning in modulating energy metabolism, either at a myocardial level or through lipolysis activation in the adipose tissue.